# Exciton dissociation in 2D layered metal-halide perovskites

Angelica Simbula [1] ✉, Luyan Wu[1], Federico Pitzalis[1], Riccardo Pau [1,2], Stefano Lai[1], Fang Liu [3], Selene Matta[1], Daniela Marongiu [1], Francesco Quochi [1], Michele Saba [1] ✉, Andrea Mura[1] & Giovanni Bongiovanni[1]

Layered 2D perovskites are making inroads as materials for photovoltaics and light emitting diodes, but their photophysics is still lively debated. Although their large exciton binding energies should hinder charge separation, significant evidence has been uncovered for an abundance of free carriers among optical excitations. Several explanations have been proposed, like exciton dissociation at grain boundaries or polaron formation, without clarifying yet if excitons form and then dissociate, or if the formation is prevented by competing relaxation processes. Here we address exciton stability in layered Ruddlesden-Popper $PEA_2PbI_4$ (PEA stands for phenethylammonium) both in form of thin film and single crystal, by resonant injection of cold excitons, whose dissociation is then probed with femtosecond differential transmission. We show the intrinsic nature of exciton dissociation in 2D layered perovskites, demonstrating that both 2D and 3D perovskites are free carrier semiconductors and their photophysics is described by a unique and universal framework.

The flexibility of hybrid metal-halide perovskites makes it possible to realize multiple crystalline configurations, including layered 2D structures[1–5]. Bidimensional confinement of optical excitations induces a blueshift of the bandgap with respect to 3D parent compounds, enhances the exciton binding energy[1,6–9] and favors symmetry breaking, Rashba spin-dependent energy splitting and ferroelectricity[10–14]. The optical absorption spectrum of layered perovskites is dominated by an intense exciton peak below the band edge, with estimated binding energies of hundreds of meV, leading to the assumption that light absorption creates bound excitons as excited states. Layered perovskites have found ample use in photovoltaics, notwithstanding the fact that bound excitons should be detrimental to efficient separation of opposite charge carriers in solar cells[15–26]. Several experiments have started to explain such apparent contradiction by presenting evidences for the presence of free carriers among the optical excitations in layered perovskites[27–35], including the observation of bimolecular radiative recombination, with rates quadratic in the concentration of optical excitations, and the transient increase of conductivity measured by THz absorption[36]. The first physical mechanism invoked to explain the abundance of free carriers has been exciton dissociation by intragap edge states at crystalline grain boundaries[27,37,38]. Such mechanism is extrinsic and related to the fact that solution-processed thin films are typically polycrystalline, with grain boundaries hosting a high density of defect edge states and dangling bonds[32,35]. Another proposed mechanism involves an intrinsic exciton dissociation, mostly explained with the formation of polarons, unbound carriers coupled to lattice deformations[9,34,35,39–44]. Such mechanism involves the rearrangement of the crystal lattice around optical excitations and lowers the energy of unbound carriers with respect to excitons[9,34,40,43–63]. The very dynamics that produces free carriers is not clear, whether optical excitation creates mostly bound excitons, that then dissociate, or the formation of unbound charges occurs in the first place and prevents the formation of stable excitons.

[1]Dipartimento di Fisica, Università degli Studi di Cagliari, Monserrato, CA I-09042, Italy. [2]Zernike Institute for Advanced Materials, University of Groningen, Nijenborgh 4, 09747 AG Groningen, The Netherlands. [3]School of Environmental Science and Engineering, Frontiers Science Center for Transformative Molecules, Shanghai Jiao Tong University, Shanghai 200240, China. ✉e-mail: angelica.simbula@unica.it; saba@unica.it

In this work we set to investigate the origin of free carriers in layered perovskites by studying the stability of excitons created by resonant excitation with negligeable excess energy. To discriminate extrinsic and intrinsic effects, in addition to thin films, we also studied single crystal samples, where the defect density is expected to be much lower and concentrated close to the surfaces[32], and generated optical excitations with two-photon absorption in the bulk of the crystal, far from any surface, and thus from any edge state. Excitons were then observed to dissociate into free carriers within 1-2 ps. We identified the intrinsic microscopic process responsible for exciton dissociation as the formation of polarons.

## Results
### Material and methods
The chosen layered perovskite for such a study was the prototypical $n = 1$ member of the Ruddlesden-Popper (RP) family $PEA_2PbI_4$, with a single slab of $(PbI_6)^{4-}$ octahedra separated by long layers of phenethylammonium (chemical formula $C_6H_5CH_2CH_2NH_3$, hereinafter abbreviated with PEA). Considering the general expression of perovskite $ABX_3$, the purest phase structure is obtained when the only cation in site A is the large-sized organic molecule, meaning that the only possible structure is constituted by single layers of octahedra ($n = 1$, with $n$ indicating the number of layers of octahedra) separated by slabs of organic molecules, as reported in Fig. 1a. Therefore, in the $n = 1$ compound, grains with higher values for $n$ cannot form, and possible extrinsic effects at the interface between different phases are avoided. The X-ray diffraction (XRD) pattern on finely ground crystals and thin films (Fig. 1b) confirmed the same crystalline structure for both samples, and the existence of a single phase. Thin films were less than 100 nm in thickness and showed an out-of-plane preferred orientation, while single crystals (SC) were produced with a thickness of around 2 microns, thin enough to allow optical transmission measurements, as necessary to probe the crystal bulk. Additionally, SC-XRD was performed on a self-standing single crystal to confirm the crystal symmetry and layered structure with refined cell parameters (see Supplementary Table 1).

The optical absorption spectrum of $PEA_2PbI_4$, shown in Fig. 1c for a thin film, was dominated at the band edge by a very well-defined exciton peak centered at 510 nm at room temperature. In RP perovskites the dielectric shielding effect from the organic cations leads to a strong quantum confinement effect in the inorganic layers, increasing with lower $n$, and with a maximum at $n = 1$. Thus, since the $n = 1$ $PEA_2PbI_4$ has the largest exciton binding energy in its RP series[4], we reasoned that it would be the most compelling perovskite to study exciton dissociation. Standard optical excitation occurs with photon

energies above the bandgap, e.g., at 430 nm wavelength. Resonant excitation at the exciton peak would cause stray light from the optical excitation to mix with probe light and photoluminescence, making it impossible to track either optical absorption or photoluminescence signals. To avoid the issue, resonant injection of excitons was instead realized by two-photon excitation with femtosecond laser pulses from a kHz regenerative amplifier. The laser wavelength could be tuned to either be in resonance with the exciton transition or to fall within the continuum above it. The small two-photon absorption cross section ensured that, even in the single crystal samples, the laser pulses were not significantly attenuated, producing a uniform excitation density across the crystal thickness.

To probe the optical excitations, ultrafast differential transmission (DT, a proxy for transient absorption) was employed[34]. The experiment was performed in three different excitation configurations: one-photon absorption with photon energy well above the bandgap (2.9 eV), two-photon absorption with twice the photon energy well above the bandgap (1.44 eV) and two-photon absorption with twice the photon energy resonant with exciton absorption (1.2 eV), as sketched in Fig. 1c. As a confirmation that indeed one and two-photon absorption was produced, the signal amplitude at time zero after excitation was measured and found to scale linear and quadratic, respectively, with laser fluence (Supplementary Fig. 1). Time resolved photoluminescence (PL) was also measured, both with a streak camera with 40 ps temporal resolution and with an optical Kerr gate achieving 300 fs time resolution.

### Ultrafast spectroscopy on $PEA_2PbI_4$ thin films
The PL spectrum from $PEA_2PbI_4$ thin films (Fig. 1c) was always resonant with the exciton absorption line and far from the band-to-band continuum; therefore, optical emission should always be ascribed to excitons, and PL could be used as a proxy to monitor exciton population in time. On the other hand, PL measurements with a streak camera collected by the very same region where the DT was probed revealed that, within the 40 ps resolution, the time decay of PL signal followed the decay of the square of the DT (see Supplementary Figs. 2, 3). Since the DT signal was proportional to the overall population, while PL was proportional to only the exciton population, such observation provided evidence that the majority of optical excitations were free carriers, while the excitons responsible for light emission were the minority species[34,64]. While such observation was just a confirmation of previous reports in literature, it was still noteworthy that, in spite of the very large exciton binding energy, at least an order of magnitude larger than thermal energy $k_BT$ at room temperature, the majority optical excitations in $PEA_2PbI_4$

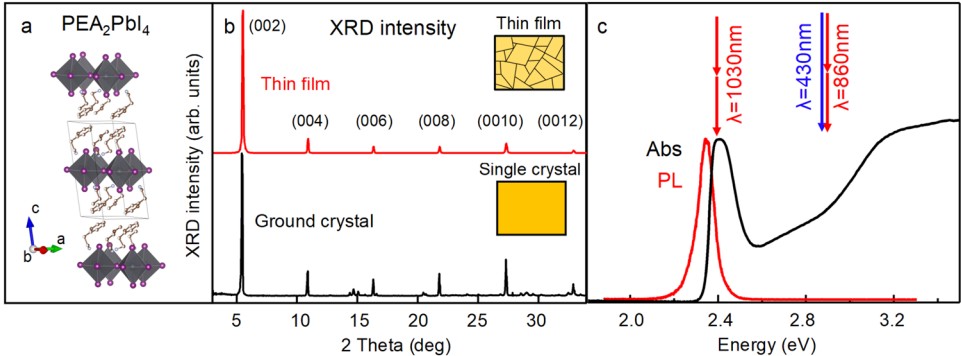

**Fig. 1 | Crystal structure and characterization. a** Sketch of the refined structure of $PEA_2PbI_4$ as obtained by SC-XRD. The unit cell is shown with a black line. **b** XRD patterns of $PEA_2PbI_4$ in the form of thin film (top) and ground crystals (bottom) demonstrating that in both cases the same $n = 1$ RP phase-pure structure was formed, with the insets showing a sketch of the morphological difference between thin films and crystals. **c** Optical absorption spectrum of $PEA_2PbI_4$ polycrystalline film measured with a UV-Vis spectrophotometer (black line) and photoluminescence spectrum (red line). The arrows show the excitation energy for resonant and non-resonant injection, with either one or two-photon absorption.

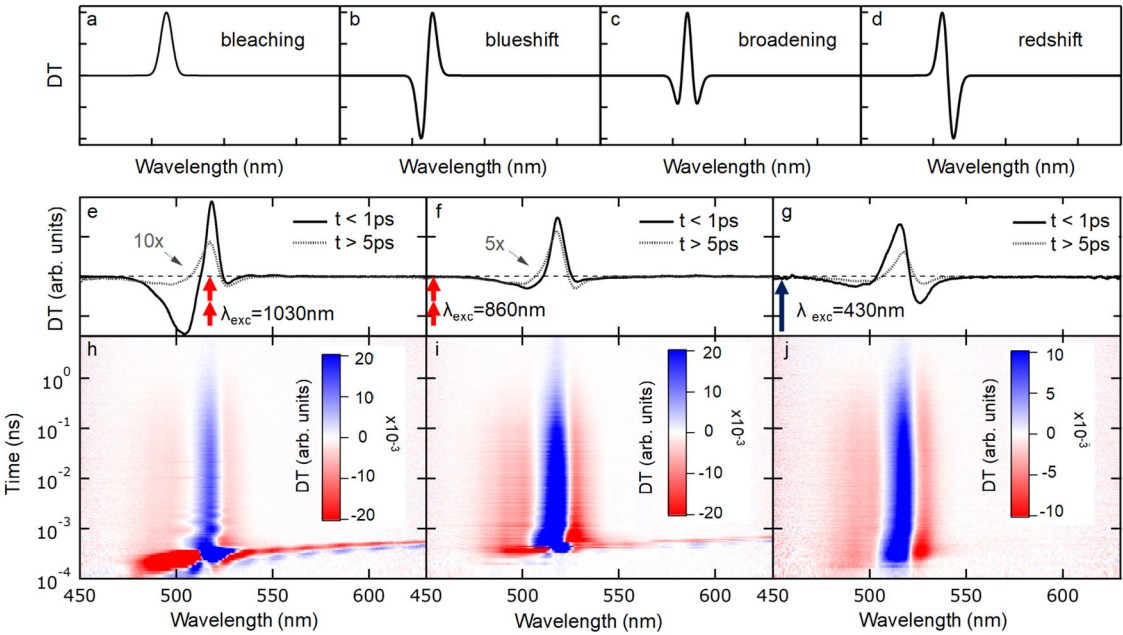

**Fig. 2 | Differential transmission measurements on thin films.** Top row (**a**–**d**): sketches of the expected DT signal, assuming a gaussian lineshape for exciton absorption, in case of bleaching, blueshift, broadening and redshift, respectively. Middle row (**e**–**g**): DT spectra measured on PEA$_2$PbI$_4$ thin-films, integrated in the first picosecond (solid line) after the arrival of the pump optical pulse and in the 5-10 ps delay window (dotted lines), for excitation wavelengths of 1030 nm (**e**, resonant two-photon), 860 nm (**f**, non-resonant, two-photon) and 430 nm (**g**, non-resonant, one photon). Bottom row (**h**–**j**): DT spectrograms from which spectra **e**–**g** were extracted, respectively, with time axis in logarithmic scale. Excitation fluences are 5 mJ/cm$^2$/pulse, 3 mJ/cm$^2$/pulse and 4 µJ/cm$^2$/pulse for 2-photon resonant, 2-photon non-resonant and 1-photon non-resonant, respectively.

were free carriers, but still light emission occurred from bound excitons.

We then analyzed the ultrafast spectral evolution of the DT signal to investigate the subpicosecond dynamics leading to such abundance of free carriers. In the presence of both excitons and free carriers, the nonlinear effects causing a DT signal could be of different origin, leading to a rich phenomenology in the DT spectra and transients[34,65–68]. Figure 2a–d represent the DT spectra outcomes that may be expected in various scenarios, assuming a gaussian exciton absorption line. Pauli exclusion prevents excitation of already occupied states and may cause a reduction in the oscillator strength of the excitonic optical transition, with a corresponding positive DT signal called bleaching; the predicted DT lineshape associated to bleaching reflects the exciton absorption spectrum and is just a gaussian (Fig. 2a). On the other hand, exciton-exciton repulsion, due to both exchange effects and Coulomb repulsion, is known to cause a blueshift in optical absorption, without significantly altering the overall oscillator strength. The DT spectrum associated to a blueshift is the first derivative of the exciton absorption line, with a negative sign (Fig. 2b). Broadening of the exciton line may result from scattering between optical excitations, mainly by free carriers, and is the analog of collisional broadening in atomic physics, although is often referred to as excitation-induced dephasing in the semiconductor community. The effect of broadening is to redistribute the oscillator strength of the optical transition from the line center, where it is lowered, to the tails, where it is increased. The predicted DT lineshape for broadening is the second derivative of the exciton absorption line (Fig. 2c). Stark effect because of trapped carriers and bandgap renormalization by hot free carriers are also known to produce a redshift of the exciton line, with an expected lineshape shown in Fig. 2d, that is the mirror image of the blueshift one.

The different excitation conditions of a PEA$_2$PbI$_4$ thin film realized some of the scenarios described above. We first analyze the case of two-photon resonant excitation and then compare it with non-resonant case. The evidence for creation of cold excitons with

resonant excitation consisted in a blueshift in the first picosecond after excitation, attributed to the strong exciton-exciton repulsion. While the blueshift was prevalent, the residual amount of bleaching could be estimated by the ratio between the spectral integrals of the relative (signed) value of the DT signal and the integral of its absolute value, i.e., the mismatch between the positive and negative lobe of the signal, which amounted to 5%. In the spectrogram in Fig. 2h oscillations were also clearly visible, with a period of one picosecond, in agreement with reports in literature[43,62,69]. A more detailed study of this phenomenon as a function of temperature and excitation wavelength, reported in Supplementary Fig. 4, suggested that the oscillations stemmed from coherent phonons in the fundamental state, triggered by a Raman process occurring during the pump pulse. After few picoseconds time elapsed, the blueshift spectrum turned into a broadening one, with much lower amplitude, an indication that excitons had transformed in a different species (Fig. 2e)[65,66]. The DT spectrum did not change anymore over the course of the entirety of the following decay. We therefore interpreted the transformation from blueshift to broadening in the DT signal as evidence for the dissociation of excitons into free carriers. Non-resonant excitation with two photons (Fig. 2f, i) produced instead a combination of broadening and blueshift. We interpreted the finding as evidence of competition between formation of excitons and free carriers during the excitation pulse duration, either because free carriers were directly created by laser excitation or because hot excitons dissociated into free carriers within a time comparable to the laser pulse duration. Non-resonant one photon excitation produced yet a different spectrum, a combination of broadening and redshift (Fig. 2g, j). A possible explanation was that one- and two-photon excitations accessed different states and produced an initial different combination of excitons and free carriers[10]. Furthermore, the one photon excitation profile across the film depth created the maximum carrier density close to the surface, making it easier for excitons to dissociate at surface trap states, and thus causing the broadening and redshift in the DT signal.

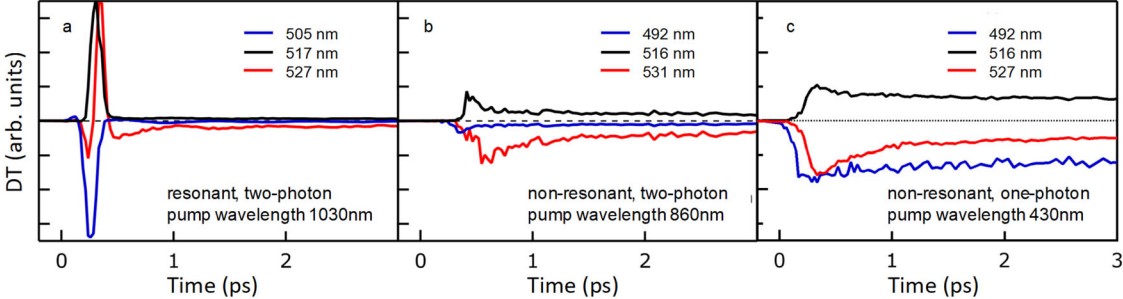

**Fig. 3 | Ultrafast time dynamics of DT spectral features.** DT signal extracted at the central peak (black lines), the long wavelength lobe (red lines) and the short wavelength lobe (blue lines) in case of: **a** resonant two-photon excitation (1030 nm in wavelength); **b** non-resonant two-photon excitation (830 nm in wavelength); **c** non-resonant one photon excitation (430 nm in wavelength). Excitation fluences are 5 mJ/cm²/pulse, 3 mJ/cm²/pulse and 4 μJ/cm²/pulse for 2-photon resonant, 2-photon non-resonant and 1-photon non-resonant, respectively.

Irrespectively of the excitation conditions, after the initial few picosecond transients, only a broadening signal remained, with similar dynamics on the nanosecond time scale. The observed spectra did not depend significantly on laser fluence (see Supplementary Fig. 5 for additional spectra at different fluences reporting the same behavior).

Figure 3 displays the time dynamics of the DT signal integrated over 5 nm wide spectral regions, the central region resonant with the exciton absorption (black lines) and the two sides (red and blue lines respectively for the long and short wavelength side lobes). Under resonant two-photon excitation (Fig. 3a) the central peak quickly decreased, as well as the short wavelength lobe; the long wavelength lobe instead switched from positive to negative when the signal turned from a blueshift into a broadening. For non-resonant excitation instead, the DT signal at the long wavelength lobe never switched sign. We therefore identify the ultrafast decay of the DT in the first picosecond and the inversion of the signal sign at the long wavelength lobe for two-photon resonant excitation as the markers for the exciton dissociation. The DT amplitude in the initial transient of resonant injection of cold excitons (blueshift signal) was much larger than that obtained under non-resonant excitation, either for broadening, redshift, or bleaching signal (Fig. 3a–c). For delays longer than a few picoseconds, when only a broadening spectrum was left, the amplitudes in absolute value of all the three signals from the central and side lobes decayed with the same law (see Supplementary Fig. 6), which we interpreted as evidence that the excited species did not change anymore.

To investigate other possible origins for the fast drop of the blueshift signal, apart from exciton dissociation, DT measurements were repeated with co-circular and counter-circular pump and probe polarizations, as reported in Supplementary Fig. 7. Measurements demonstrated no significant differences between the two cases, excluding significant contributions to our observations from polarization-related effects, such as biexcitons and exciton-exciton correlation, that have been observed elsewhere at low temperatures[70].

The main message from Fig. 3 was that resonantly injected excitons in thin film PEA$_2$PbI$_4$ were unstable and they dissociate into free carriers within a few picoseconds. When excitation was non-resonant, relaxation to band bottom, exciton formation and dissociation all competed with similar time scales, so that a majority of excitons may have never formed.

## Ultrafast spectroscopy on PEA$_2$PbI$_4$ crystals

We then investigated the origin of the exciton dissociation process, whether it was intrinsic or extrinsic. To this aim, single crystal samples were studied, since two-photon resonant excitation could create cold excitons within the crystal bulk, far away from any surface or grain boundary. PL spectra from the crystal were redshifted with respect to the thin film counterpart (solid and dashed green lines in Fig. 4a) since the PL emitted from the crystal bulk underwent strong re-absorption, due to the limited Stokes shift.

To access the bulk of the crystal, differential transmission was employed (while differential reflection would only probe a depth of few tens of nm from the surface), but the peak absorption coefficient exceeded $10^5$ cm$^{-1}$, so that the light transmitted by the crystal in resonance with the excitonic peak was negligible. Therefore, as shown in Fig. 4a, the DT signal was not accessible in a spectral region corresponding to the peak of the excitonic transition, while only the sidebands could be detected clearly. The differential transmission signal in the "blanked out" wavelength range was close to zero (instead of being noise-like signal, as would be expected in case of very low transmission), which was attributed to the detection of residual scattered probe light that did not cross the region of the crystal excited by pump pulses. The most relevant experiment to understand exciton dissociation was the resonant injection of cold excitons with two-photon excitation (1030 nm pump wavelength), whose results are reported in Fig. 4. Right after resonant injection of cold excitons by two-photon absorption, an asymmetric spectrum was observed with a mostly positive DT amplitude at long wavelengths and a negative one at short wavelengths, as expected for a blueshift (black solid line Fig. 4a, t < 1 ps). Apart from the central part of the spectrum, which was not observable, such a spectrum was compatible with the blueshift one observed in the thin films under resonant two-photon excitation. Within few picoseconds, the spectrum transformed in just two negative sidebands (black solid line Fig. 4a, t > 5 ps), compatible with the broadening signal observed in thin films.

The time dynamics of the high and low energy lobes, shown in Fig. 4b, were also similar to what reported for thin films (Fig. 3a). The short wavelength lobe quickly decayed, while the long wavelength lobe switched sign, from positive to negative, as observed in the thin film for the transition from blueshift to broadening associated to exciton dissociation. On longer time scales, the two sidebands evolved with the same decay, as the spectrum did not change anymore, with the signal amplitude monitoring the overall population. The same consistent phenomenology was observed for different excitation densities, with signal amplitudes varying with the excitation laser fluence, but reproducing the same qualitative evolution, both in thin films and thin crystals (see Supplementary Figs. 8, 9). Non-resonant two-photon excitation produced similar results, but with a less pronounced initial blueshift, like in thin film samples. The relevant spectrograms are reported in Supplementary Fig. 10.

PL signals from the thin crystals excited with two-photon absorption are shown in Fig. 4c in the first few picoseconds after excitation. For both resonant and non-resonant excitation, the dynamics were characterized by a non-instantaneous response, with a rise time of the order of 1 ps, significantly longer than the 300 fs resolution of the setup. Since two-photon excitation created excitons

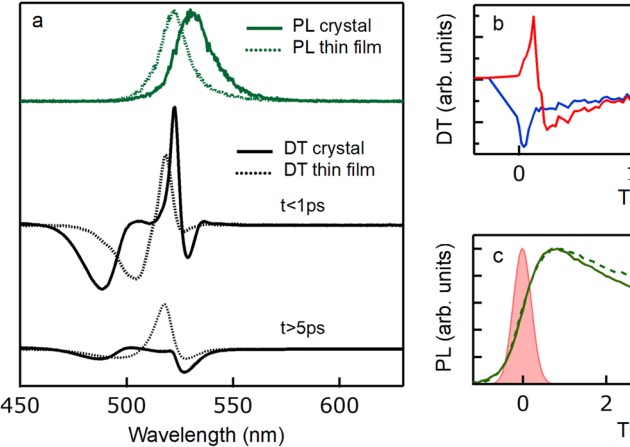

**Fig. 4 | Thin film and thin crystal comparison. a** On top: photoluminescence spectra for crystal and thin film samples (green solid and dashed lines, respectively), with 830 nm excitation for crystal (non-resonant two-photon) and 430 nm for thin film. Center and bottom: DT spectra under resonant two-photon excitation (1030 nm) from the single crystal (solid black line) and thin film (dashed black line) PEA$_2$PbI$_4$ samples right after laser excitation (t < 1 ps, center), and at longer delays (t > 5 ps, bottom) when only broadening is left. PL and DT spectra are vertically offset for clarity. Due to the crystal being much thicker than the thin film, the DT signal is more intense, leading to an improved signal-to-noise ratio. **b** Amplitude of the DT signal from the crystal sample in the short and long wavelength lobes (blue and red, respectively) as a function of time in the first 3 ps time window after pulsed excitation with a fluence of 6 mJ/cm$^2$/pulse. **c** Ultrafast PL measured with a Kerr medium optical gate setup under two-photon excitation, both resonant (1010 nm) and non-resonant (860 nm) with the exciton transition, reported as solid and dashed green lines, respectively. The pink shaded area represents the instrument response function, highlighting that the PL rise time is significantly longer than the time resolution of the setup.

that were optically dark because of spin selection rules, we interpreted the rise time of the PL signal as the time needed for a spin flip process to transform optical excitations from dark to bright. Such spin flip time was comparable to the exciton dissociation time retrieved from DT measurements, thus PL could not be used to extract the dynamics of exciton dissociation.

All experiments were repeated at cryogenic temperatures, particularly 80 K and 150 K, to investigate the influence of thermal energy on exciton dissociation. Results at low temperature are reported in Supplementary Fig. 11, and were analogous to what observed at room temperature. Remarkably, the dissociation occurred also at cryogenic temperatures.

## Discussion

The observation of exciton dissociation in a PEA$_2$PbI$_4$ single crystal excited by two-photon absorption excluded that the microscopic origin can be entirely traced back to edge states. Some intrinsic phenomenon had to be invoked to reduce the energy difference between bound excitons and free carriers with respect to the exciton binding energy observed in absorption. Based on several evidence already reported in literature, we attributed exciton dissociation to the formation of polarons, i.e., to the deformation of the crystal lattice that adapts to the presence of optical excitations[34,39-43]. Even though exciton binding energy in this material was above 200 meV, thus largely exceeding thermal energy at room temperature, their dissociation could be explained with the fact that polaron stabilization energy, i.e., the energy gained through lattice relaxation in presence of optical excitations, was comparable to the exciton binding energy[35]. Therefore, an equilibrium condition could be established, according to Saha equation, with an equilibrium constant depending on the difference between exciton binding energy and polaron relaxation energy[34].

Polarons were not the optically active states created upon optical excitations, a fact that explains the lack of significant bleaching of the exciton transition in DT experiments. The main effect of unbound polarons on the DT spectrum was the broadening of the exciton line, which was present even at long time delays after the excitation and at any fluence, meaning that polarons were efficient in scattering with excitons. The initial blueshift signal coming from the pure exciton gas

was much more intense than the signal amplitude when only broadening signal, dominated by polaron-exciton scattering, remained[65]. The amplitude of the fast initial decay corresponding to polaron formation depended on the excitation conditions: it was the highest for resonant injection of a pure exciton gas, either at cryogenic or room temperature, while it decreased for non-resonant excitation, since free carriers were directly created upon optical absorption and gave rise to polarons without the intermediate step of forming bound excitons. The initial ultrafast dynamics of DT signal at room temperature was comparable to what was measured at cryogenic temperature, as reported in Supplementary Fig. 12, suggesting that polaronic deformation was not noticeably temperature-dependent in the explored temperature range (down to 77 K).

To check if the equilibrium between excitons and polarons was established, we considered that the DT bleaching signal monitored the overall excited state density $n$, which for practical purposes coincided with the polaron density $n_p$, while the PL signal was proportional to the exciton density, PL $\propto n_X$[34]. The decay rates of $d(PL)/dt \propto \dot{n}_X$ and $d(DT)/dt \propto \dot{n}_p$ as a function of the signal amplitudes PL $\propto n_X$ and DT $\propto n_p$ represented therefore the power laws with exponents corresponding to the order of the recombination kinetics for excitons and polarons, respectively.

In thermodynamic equilibrium the exciton population is expected to be proportional to the square of the polaron population according to relation $n_p^2 = n_{eq} n_X$ ($n_{eq}$ being the equilibrium constant), and therefore the two power laws should be linked to each other[34]. Specifically, a decay with power law $\alpha$ for polarons, i.e. $\dot{n}_p \propto n_p^\alpha$, produces a power law $\frac{\alpha+1}{2}$ for exciton decay because $\dot{n}_X = \frac{2n_p\dot{n}_p}{n_{eq}} \propto n_p^{\alpha+1} \propto n_X^{\frac{\alpha+1}{2}}$, which comes just from the time derivative of the equilibrium equation. In the case of a polaron majority and a dominant bimolecular decay of polarons, the exponent of power law for polaron decay is expected to be $\alpha = 2$, with a matching exponent $\frac{\alpha+1}{2} = \frac{3}{2}$ for minority excitons.

Figure 5 reports the log-log plots of $d(PL)/dt$ vs PL and $d(DT)/dt$ vs DT for the single crystal under two-photon excitation (similar results were obtained under different excitation conditions and also in thin films, as shown in Supplementary Fig. 13). The observed matching

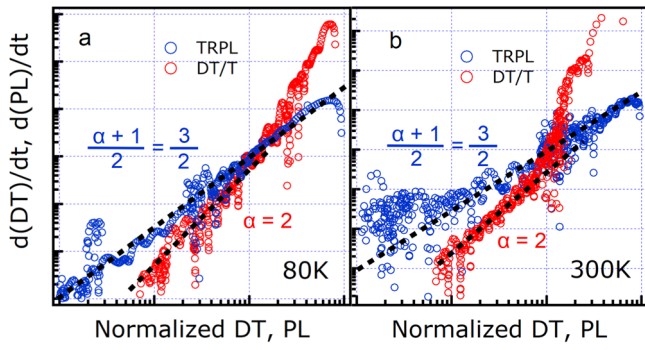

**Fig. 5 | Power laws for DT and PL.** Power laws for the absolute values of time derivatives of differential transmission (DT/T), blue circles, and time-resolved photoluminescence (TRPL), red circles, signals as a function of the signal amplitudes themselves, extracted from measurements at temperatures of 80 K (**a**) and 300 K (**b**) on a PEA$_2$PbI$_4$ single crystal under two-photon resonant excitation. Black dashed lines are a guide to the eye for slopes 3/2 (superimposed with DT/T blue circles) and 2 (superimposed with TRPL red circles).

power laws for DT and PL constituted clear evidence that, at 300 K as well as at 80 K, an equilibrium was established between a majority of unbound polarons and a minority of bright excitons, the latter ones being responsible for photoluminescence emission. The power laws reported in Fig. 5 allowed us to identify the dominant decay process as the monomolecular decay of excitons: in equilibrium, such a decay corresponded to a bimolecular depletion of polarons, with an emission process at equilibrium that can be schematized as $p^+ + p^- \leftrightarrows X \rightarrow \hbar\omega_X$.

The picture of carriers formation and equilibrium dynamics emerging from the analysis of the decays' power laws is summarized in Fig. 6: two photon absorption resonantly injected bound excitons (step 1), then exciton dissociated into polarons (step 2), until equilibrium condition was reached between exciton dissociation and polaron pairing into excitons (step 3); light emission occurred when excitons recombined radiatively (step 4).

Polaron formation appeared in Fig. 5 as a deviation from the DT power law occurring when the signal was highest, i.e., in the initial picoseconds after optical excitation. The decay of the DT signal due to polaron formation followed a power law with exponent ≈5, much faster than the bimolecular decay with exponent 2 in equilibrium. Such fast transient occurred out of equilibrium and therefore the decay rate was not uniquely determined by the concentration of optical excitations. On the contrary, the initial transient was observed for all employed laser fluences and thus for all initial exciton concentrations. Finally, it is likely that, before and after equilibrium between unbound polarons and excitons was established, the exciton wavefunction was significantly altered by lattice deformations, to the point of forming exciton-polarons, as reported in literature[9,43].

Alternate phenomena to explain the observed DT and PL power laws, such as exciton-exciton annihilation and Auger recombination, that have been reported in literature, could be ruled out. Exciton-exciton annihilation is a bimolecular process that reduces the exciton density, but does not produce free carriers, and thus would cause the same power law for both PL and DT signals. Auger recombination would instead only produce a bimolecular power law for minority excitons, with a corresponding recombination dynamic of order $\alpha = 3$ for polarons (with $\frac{\alpha+1}{2} = 2$ for excitons) and is thus not compatible with the observed kinetics.

Because polarons are dark, all evidence collected from ultrafast optical spectroscopy about their formation and dynamics was necessarily indirect and could not access distinctive features, such as their stabilization energy, untangle electron and hole polarons, and quantify the lattice displacements involved. Therefore, a crucial need appears

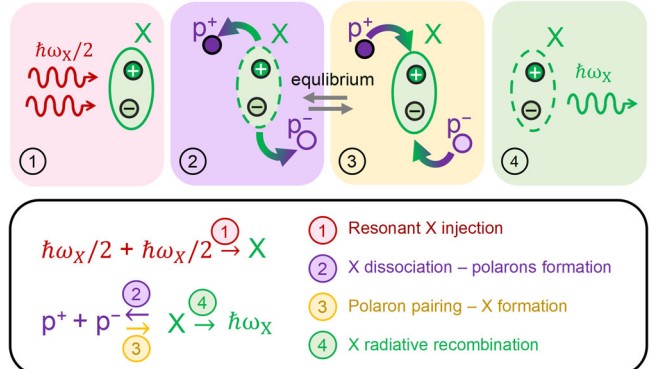

**Fig. 6 | Schematic representation of exciton dissociation into free carriers and equilibrium dynamics.** In the upper panels, four sequential steps are shown, representative of the dynamics triggered by resonant two-photon pulsed laser excitation: **1** Two-photon resonant excitation resulting in the direct injection of excitons (X); **2** Dissociation of excitons into unbound dark polaron pairs p$^+$ and p$^-$; **3** Pairing of dark polaron pairs into bright excitons; **4** Exciton radiative recombination. Between steps 2 and 3, an equilibrium condition is established between a majority population of polaron pairs and minority one of bound excitons. In the lower panel, each of the steps is named and represented by the corresponding reaction scheme.

for future research to access directly polaron deformation through measurements sensitive to the lattice displacements, such as ultrafast X-ray or electron diffraction[44,71].

In conclusion, we designed an ultrafast spectroscopy experiment to inject cold excitons in PEA$_2$PbI$_4$ layered perovskites by two-photon absorption. Spontaneous exciton dissociation was observed on a picosecond time scale, which we interpreted as formation of polarons. As a consequence, a single photophysical framework described all hybrid perovskites, based on the equilibrium between a majority of dark polarons and a minority of bright excitons. The findings presented here establish that, notwithstanding their large exciton binding energies, layered 2D perovskites are free carrier semiconductors like their 3D counterparts and vindicate their use in solar cells, where their superior stability is not curtailed, as commonly assumed, by a voltage penalty to dissociate bound excitons.

## Methods
### Materials preparation
Glass substrates were washed with detergent, cleaned by sequential ultra-sonication in detergent, ID water, acetone, and isopropanol. Each ultra-sonication process lasted for 15 min. Cleaned substrates were treated by an oxygen plasma asher for 3 min after drying with N$_2$ stream.

2D PEA$_2$PbI$_4$ single sheet crystals were synthesized from solution of PbI$_2$ (0.25 M) and PEAI (0.25 M) dissolved in 5 mL mixture of HI and H$_3$PO$_2$ (10:1 vol/vol) at 130 °C for 2 h. Then the clear, yellow solution was cooled down to 70 °C[72]. Supernatant solution was collected after filtering at 70 °C. 10 μL of supernatant solution were dropped on the clean glass substrate at room temperature in a fume hood. Then many single sheet crystals started to grow within few seconds on the glass substrate. At last, chlorobenzene was used to wash them three times.

The 2D PEA$_2$PbI$_4$ precursor solution was prepared by dissolving PEAI (0.7 mmol), PbI$_2$ (0.35 mmol) in 1 mL Dimethyl sulfoxide (DMSO). 80 μL of precursor solution were dropped on an 80 °C pre-heated substrate and spun at 3000 rpm for 30 s. The films were annealed at 100 °C for 10 min after spin-coating. Before measurement, films were coated with two drops of polymethyl methacrylate (PMMA) solution 50 mg/ml in Chlorobenzene at 6000 rpm for 30 s. All the film process was operated inside a N$_2$-filled glovebox. More details about the choice

of the solvent and the effect of PMMA deposition can be found in Supplementary Fig. 14.

## Materials characterization

The X-ray diffraction patterns of the films were performed with a Bruker Advance X-ray diffractometer equipped with a Goebel mirror in a theta/2theta geometry and recorded with 0.025° step size and 0.5 s/step. The single crystals were finely ground before the measurements with the same instrument in a Bragg-Brentano geometry.

UV–vis optical absorption of HP films was measured with a dual-beam spectrophotometer equipped with an integrating sphere accessory (Agilent Technologies Cary 5000 UV–vis–NIR) to collect both diffused transmission and reflection.

## Ultrafast tandem spectroscopy setup

The source of ultrafast laser light is a Ti:sapphire regenerative amplifier (Coherent Libra) delivering 100-fs long pulses, 794 nm in wavelength, up to 4 mJ in energy, 1 kHz repetition rate. The conversion from fundamental wavelength to different excitation energies was achieved by means of an optical parametric amplifier (Topas 800 from Light Conversion) equipped with nonlinear crystals. The DT spectrometer (Helios from Ultrafast Systems) measured individual probe and reference in differential configuration, collecting spectra with CMOS spectrometers (1 nm spectral resolution). Homemade modification of Helios system allowed to collect luminescence from the same excitation spot. For long time range and time resolution down to 40 ps, the PL signal is dispersed with a grating spectrometer (Acton2300i) and then detected with streak camera (Hamamatsu C10910). For sub-picosecond temporal resolution, a Kerr gate setup was realized, employing nanoparticle linear film polarizers and focussing the PL signal on a 1 mm quartz, where an intense gate pulse, 800 nm in wavelength, was overlapped to generate the Kerr gate. The gated PL was dispersed in a grating spectrometer (Acton2300i) and then detected with CCD camera (Andor Newton); the temporal resolution was 300 fs. The acquisition of PL and DT was done sequentially, and the white light probe was blocked while measuring PL. Low temperature measurements were performed keeping the sample high-vacuum using a cryostat (Janis ST-500) and liquid nitrogen.

## Data availability

The datasets generated and analysed during the current study are available from the corresponding authors upon request.

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

## Acknowledgements

All the authors acknowledge CeSAR—Centro Servizi di Ateneo per la Ricerca—at the Università degli Studi di Cagliari and thank Dr. M. Marceddu and Dr. E. Podda for technical assistance. This work was funded by Fondazione di Sardegna through project 2F20000210007

"Perovskite materials for photovoltaics" and project F73C22001160007 "Single crystal hybrid perovskite thin films for optoelectronics". A.S. was supported by PON "Ricerca e Innovazione" 2014–2020—Fondo sociale europeo, Attraction and International Mobility—Codice AIM1809115 Num. Attività 2, Linea 2.1. Funding is acknowledged under the National Recovery and Resilience Plan (NRRP), Mission 4 Component 2 Investment 1.3 - Call for tender No. 1561 of 11.10.2022 of Ministero dell'Università e della Ricerca (MUR); funded by the European Union – NextGenerationEU, Project code PE0000021, Concession Decree No. 1561 of 11.10.2022 adopted by Ministero dell'Università e della Ricerca (MUR), CUP – F53C220007700077, according to attachment E of Decree No. 1561/ 2022, Project title "Network 4 Energy Sustainable Transition – NEST".

## Author contributions

A.S., R.P., F.P., A.M., F.Q., M.S. developed the ultrafast spectroscopy setup and carried out the optical measurements. A.S., F.P. and M.S. performed the data analysis. L.W., F.L., S.M., D.M., S.L. carried out the material synthesis, samples preparation. L.W., D.M., S.L. performed XRD and absorption measurements. A.S., M.S., F.Q and G.B. wrote the manuscript, and all the authors contributed to the discussion and approved the final version.

## Competing interests

The authors declare no competing interests.
