## [Peer Review File · Nature Communications]

Exciton dissociation in 2D layered metal-halide perovskitesReviewers' Comments:

Reviewer #1:

Remarks to the Author:

In this manuscript, the authors utilize 2D Ruddlesden-Popper halide perovskite to discuss the exciton stability. The ultrafast spectroscopy experiment based on differential transmission and photoluminescence is demonstrated. Spontaneous exciton splitting is observed and interpreted as a generation of polarons. The authors conclude that 2D halide perovskite is a free carrier semiconductor like their 3D structures, although with larger exciton binding energy. The experiment and discussion in this manuscript are insightful and convincing. The authors are recommended to address the following comments to further improve the quality of this work.

1. In line 42, edge states and dangling bonds could also exist on single-crystal surfaces. (Na Quan, Li et al. "Edge stabilization in reduced-dimensional perovskites." Nature communications 11.1 (2020): 1-9.)
2. In line 61, page 3, it says, "PEA is the only cation." But lead also serves as a cation in the structure. In the following sentence, "in the absence of a second shorter organic cation," it is not only organic cation that could occupy the position, like PEA₂CsPb₂Br₇. It is suggested that the authors make the sentence clearer and more accurate.
3. The XRD pattern in Fig 1a is usually not sufficient to prove the crystalline structure of 2D halide perovskite for its layered structure. It is recommended that the authors take small-angle X-ray scattering images to discuss its structures better. More information will be revealed.
4. In Figure 1b, the XRD of the polycrystalline thin film had fewer peaks than the single crystalline sample. However, there should be more than one orientation in polycrystalline samples, which contributes to more XRD peaks. The authors should explain why there are more peaks in the single crystalline sample.
5. The authors should distinguish whether their thin films are single-crystalline or polycrystalline in the main text.
6. It would be more straightforward for bandgap discussion if the authors could integrate photoluminescence and absorption into one figure in Fig 1c.
7. In line 70, page 3, the authors need a reference to support that PEA₂PbI₄ has the largest exciton binding energy of the whole RP series with PEA.
8. In line 121, page 6, how and by what data could the authors derive a "5%" bleaching?
9. In line 122, the authors should briefly describe the phenomenon of coherent oscillations and why it appears here. And the authors use citations 40 and 63 to explain the coherent oscillations. The authors also claimed that the appearance of polarons might come from the deformation of the crystal lattice in line 250. The authors should further explain the relationship between them.
10. Compared with Fig 2e-g, why does DT signal in Fig 4a look so smooth, without any noise? Is there any data processing within it? If so, the authors should clarify it.
11. In line 143, the authors said the one-photon excitation could create the maximum carrier density of carriers and make it easier for opposite charge carriers to split at the surface trap states. The "opposite charge carriers" may be ambiguous because the authors said the carriers could split. And the "split" is only used to describe excitons instead of carriers in this manuscript. In the reviewer's opinion, the authors are trying to express that free carriers were generated and combined into excitons. Those excitons close to the surface tend to split at the surface trap states, which causes the broadening and redshift in the DT signal.
12. In line 257, the authors claimed that the fast initial decay corresponds to polaron formation. Because the temperature would influence on the speed of lattice deformation, the initial decay will also be influenced by temperature. The authors should supplement the experiments by comparing the initial decay with different temperatures.
13. Do the authors attempt to improve the quality of the polycrystalline thin film? According to the methods part, utilizing DMSO to dissolve the precursors is rare for iodide 2D perovskite. Meanwhile, the spin coating and annealing parameters are also critical to the film quality. Comparing single crystals and polycrystals is unfair when one is in poor condition. Also, the PMMA treatment could potentially

affect the characterization. How did the authors evaluate the impacts?

14. How does halide perovskite superlattice (Nature volume 608, pages 317–323 (2022)) compare with bulk single crystals and polycrystalline thin films in the exciton stability?

15. The image quality of the main text should be improved.

Reviewer #2:

Remarks to the Author:

The manuscript describes the results of the pump probe measurements of the 2D perovskites, namely PEA₂PbI₄ in n=1 Ruddlesden-Popper phase, both in single crystal and thin film form. Resonant excitation was used to create cold excitons. The main claim of the manuscript is exciton dissociation in 2D Ruddlesden-Popper layered perovskites, which would make them similar to their 3D counterparts. This is based on the observed splitting into free carriers within 1-2 ps as a consequence of the formation of polarons.

The experimental data are interesting, however I do not find that they support the final claim of the manuscript.

1. Why the splitting of the exciton signal in pump probe cannot be attributed to the formation of biexciton? For this polarization resolved measurements should be performed.
2. What are physical evidences of the polaron formation? For this either PL on a very short time scale should be measured (ACS Nano 2022, 16, 12, 21259–21265) or enhancement of the effective mass of the carrier, a characteristic sign of the polaron should be provided.
3. At the same time I do not understand the physical process proposed by authors, is it polaron exciton formation? or is it dissociation and then carrier polaron formation? To which modes in each scenario there sill coupling. This has to be quantified.
4. Temperature dependent measurements should be provided, as coupling with the lattice should have a characteristic temperature dependence.
5. What would be the mechanism of the dissociation of exciton with binding energy of few hundred meV? Why would polaron dissociate the exciton?
6. Just bimolecular decay si not sufficient to claim charge separation, as in 2D TMDS, where we have strongly bound exciton and still Auger effect is observed.

Before providing this extra data and answer to above question, the manuscript is not suitable for publication.

Reviewer #3:

Remarks to the Author:

The manuscript "Exciton splitting in 2D layered metal-halide perovskites" presents a smart experimental study of dynamics of dissociation of bound excitons in a two-dimensional (Phenethylammonium)₂PbI₄ perovskite using time-resolved femtosecond differential transmission techniques. The results indicating their dissociation into free carriers within 1-2 ps can be important for understanding of nature of exciton dynamics in 2D Ruddlesden-Popper layered perovskites and implementation of the materials into photovoltaic devices.

Unfortunately, the authors use some jargon in the manuscript, such as "exciton splitting" and "excitons form and then split" which obscures the physics of the processes taking place. It is difficult

to imagine that exciton split something or be spatially split in the material. Probably, a splitting of their energy states is meant. In this case, it would be better if the authors used more precise terminology. It also remains unclear from the manuscript the value of this splitting. Does it correspond to the wavelength distance between the maximum and the minima in Fig. 2e? In order for the reader to better perceive the processes that the authors use to explain the experimental results, it is absolutely necessary to give an energy scheme with an indication of transitions on it at different excitations used by the authors in the experiment.

Other comments:

Specifying $n = 1$ in the abstract is not clear and requires an explanation. "PEA": using abbreviation in the abstract is not a good style.

Section Material and methods, first paragraph:

" $n = 1$ " is still puzzling, what is n here? A physical value or a number? If latter, in what sequence? Can authors characterize the phase they mention here by more informative terms than " $n = 1$ "?

Resolution of Fig. 1a needs to be improved.

Point-by-point response to the reviewers' comments

Reviewer #1 (Remarks to the Author):

In this manuscript, the authors utilize 2D Ruddlesden-Popper halide perovskite to discuss the exciton stability. The ultrafast spectroscopy experiment based on differential transmission and photoluminescence is demonstrated. Spontaneous exciton splitting is observed and interpreted as a generation of polarons. The authors conclude that 2D halide perovskite is a free carrier semiconductor like their 3D structures, although with larger exciton binding energy. The experiment and discussion in this manuscript are insightful and convincing. The authors are recommended to address the following comments to further improve the quality of this work.

1. In line 42, edge states and dangling bonds could also exist on single-crystal surfaces. (Na Quan, Li et al. "Edge stabilization in reduced-dimensional perovskites." Nature communications 11.1 (2020): 1-9.)

We thank the Reviewer for the comment and agree that our statement could be made more precise. In this specific case, we mean that the concentration of surface defects in the polycrystalline film is much higher than in the single crystal due to the presence of grain boundaries [Blancon et al., Science 355,1288–1292 (2017)]. The assumption is corroborated by experimental data (Fig. S6, Supporting Information): the long-lived TA signal at cryogenic temperature, attributed to temperature-activated trapping, is clearly visible only in the case of poly-crystalline thin film, while it has negligible amplitude in the case of single crystal. We have modified and extended the manuscript to clarify our statement and motivate our assumption.

2. In line 61, page 3, it says, "PEA is the only cation." But lead also serves as a cation in the structure. In the following sentence, "in the absence of a second shorter organic cation," it is not only organic cation that could occupy the position, like PEA₂CsPb₂Br₇. It is suggested that the authors make the sentence clearer and more accurate.

We appreciate the carefulness of the Referee: we now added in the manuscript a more precise description of layered perovskite structures and clarified what was the actual meaning of this sentence, that was not accurate enough.

3. The XRD pattern in Fig 1a is usually not sufficient to prove the crystalline structure of 2D halide perovskite for its layered structure. It is recommended that the authors take small-angle X-ray scattering images to discuss its structures better. More information will be revealed.

We thank the referee for the suggestion. In the original manuscript only the XRD powder pattern was recorded and compared to literature, since the PEA₂PbI₄ perovskite has a well known crystal structure and several cif files have been already deposited in the main database. Nevertheless, the authors have considered the reviewer's suggestions to characterize the structure more in depth and performed single crystal diffraction on a small PEA₂PbI₄ crystal with a Bruker D8 Venture diffractometer equipped with Incoated microsource (Mo $K\alpha$, $\lambda = 0.71073 \text{ \AA}$) and a PHOTON II detector. A suitable crystal of $0.05 \times 0.13 \times 0.15 \text{ mm}^3$ was selected and mounted on a MiTeGen loop (50 μm in diameter). Two sets of ω scans (12 frames each, $0.5^\circ/\text{frame}$) were collected using a detector to sample distance of 50 mm and 5 s/frame. A total of 113 reflections with $I/\sigma \geq 20$ were found and were indexed with the Fast Fourier Transform Method. The initial unit cell parameters were then refined and the Bravais lattice determined using the APEX 3 plugin. The measurements confirm the symmetry and structure as triclinic with space group P-1 and the following lattice parameters: $a = 8.680 \text{ \AA}$, $b = 8.682 \text{ \AA}$, $c = 16.418 \text{ \AA}$ and $\alpha = 94.512^\circ$, $\beta = 100.618^\circ$, $\gamma = 90.559^\circ$, with disordered cations. Our results are in agreement with a crystal

structure deposited on the CCDC with refcode BARHOU for the same layered perovskite (Nano Res. (2017), 10, 2117, DOI: 10.1007/s12274-016-1401-6). Fig. 1a has been updated with the new sketch of the structure from experimental data. Table with refined cell parameters has been added in the SI.

4. In Figure 1b, the XRD of the polycrystalline thin film had fewer peaks than the single crystalline sample. However, there should be more than one orientation in polycrystalline samples, which contributes to more XRD peaks. The authors should explain why there are more peaks in the single crystalline sample.

The comment of the referee is correct: XRD patterns on single crystals should have fewer peaks than the polycrystalline one. However, the XRD pattern shown in the manuscript is obtained from a finely ground crystal which is analogous to a polycrystalline powder. The new single crystal diffraction measurements mentioned at point n.3 confirm as much. On the other hand, the polycrystalline film has a preferred orientation out of plane along the 00l direction while it is reasonably randomly oriented in-plane, for this reason, in theta-2theta geometry it is possible to see only the 00l peaks. In the modified manuscript we now include a full discussion of the issue.

5. The authors should distinguish whether their thin films are single-crystalline or polycrystalline in the main text.

We thank the referee for the question, that induced us to improve the “visualization” of the samples. While thin films are polycrystalline, they grow with a spontaneous preferred orientation along the 001 direction during the spin coating process. The main text has been modified with a more detailed description, as well as small insets in Fig. 1b to help the reader visualize the two different kind of samples that were studied in this work.

6. It would be more straightforward for bandgap discussion if the authors could integrate photoluminescence and absorption into one figure in Fig 1c.

We thank the referee for the suggestion, the Figure has been modified by integrating absorption with PL spectrum in Fig. 1c, as proposed by the reviewer.

7. In line 70, page 3, the authors need a reference to support that PEA₂PbI₄ has the largest exciton binding energy of the whole RP series with PEA.

We thank the referee for the careful observation, and agree that this statement needs a supporting reference. We also realized the sentence was misleading, since the n=1 PEA₂PbI₄ has the largest exciton binding energy in its own RP series, thus we modified the text and added a reference to the paper [Gao, X. et al. *Advanced Science* **6**, 1900941 (2019)] where a comprehensive list of layered perovskite in the Ruddlesden-Popper phase is provided, along with their bandgaps and exciton binding energies. We inserted a brief discussion illustrating that, being quantum confinement strongest for the n=1 layered compound, it is natural that also the exciton binding energy has the largest value.

8. In line 121, page 6, how and by what data could the authors derive a “5%” bleaching?

The fraction is calculated as the ratio between the spectral integrals of the absolute value of the DT signal and its relative (signed) value; as such, it measures the mismatch between the positive and negative parts of the signal. Pure bleaching would generate 100% positive signal, pure broadening would generate zero mismatch between positive and negative signal, since broadening does not alter the overall oscillator strength of the optical transition. We recognize that the number was quoted without enough explanation, which we have now explicitly added to provide context.

9. In line 122, the authors should briefly describe the phenomenon of coherent oscillations and why it appears here. And the authors use citations 40 and 63 to explain the coherent oscillations. The authors also claimed that the appearance of polarons might come from the deformation of the crystal lattice in line 250. The authors should further explain the relationship between them.

We thank the referee for carefully reading and pointing out this ambiguity, as we figured out that this needed to be emphasised. In line 122 we are referring to the phenomenon of ground state coherent phonon oscillations, as a result of lattice oscillations in the fundamental state triggered by a Raman process occurring during the pump pulse. In order to clarify if the observed oscillation is indeed coming from ground state coherent phonon oscillations, we performed DT measurements at different excitation photon energies, in resonance with the exciton peak, below and above it, with one and two photon excitation. The results are reported in Fig. S2, showing picosecond-scale oscillations at different excitation wavelengths, and temperature dependent behaviour with 2-photons resonant excitation. It is evident that in the resonant case oscillation amplitude is amplified and damping is much slower, while their period and phase are not depending on excitation energy. Furthermore, the observed period (around 1ps) corresponds to a well-known Raman mode (30 cm⁻¹) of PEA₂PbI₄, providing further confirmation of the ground-state nature of coherent oscillations.

We have now inserted a discussion to clarify that polaron formation is an excited-state phenomenon caused by lattice deformation in the presence of optical excitations, while the observed coherent oscillations are a ground-state phenomenon, caused by the Raman excitation of the lattice without a population in the excited state.

10. Compared with Fig 2e-g, why does DT signal in Fig 4a look so smooth, without any noise? Is there any data processing within it? If so, the authors should clarify it.

Thanks to the observation of the Referee, we noticed that there can be a minor difference in the background noise, but none of the reported signal was subjected to smoothing or data processing. The difference in signal-to-noise ratio (SNR) can be attributed to higher thickness of the crystal with respect to that of thin film, which translates, through Lambert-Beer law, in a higher absorption. The thickness of the crystal is in fact approximately one order of magnitude higher than that of the thin-film, leading to a higher DT signal and a lower SNR. We have now inserted a comment to briefly clarify this point.

11. In line 143, the authors said the one-photon excitation could create the maximum carrier density of carriers and make it easier for opposite charge carriers to split at the surface trap states. The “opposite charge carriers” may be ambiguous because the authors said the carriers could split. And the “split” is only used to describe excitons instead of carriers in this manuscript. In the reviewer’s opinion, the authors are trying to express that free carriers were generated and combined into excitons. Those excitons close to the surface tend to split at the surface trap states, which causes the broadening and redshift in the DT signal.

We are grateful for this suggestion by the Referee, which was remarked by Referee 3 as well. Having realized that our phrasing may be ambiguous and in conflict with other terminology, we decided to edit the manuscript by replacing expressions as “exciton splitting” or “carrier splitting” with “exciton dissociation”, even in the title. We believe that this modification is greatly improving the clarity of the whole manuscript and of the scientific claims.

12. In line 257, the authors claimed that the fast initial decay corresponds to polaron formation. Because the temperature would influence on the speed of lattice deformation, the initial decay will also be influenced by temperature. The authors should supplement the experiments by comparing the initial decay with different temperatures.

We thank the Referee for raising this point and accept the suggestion of additional measurements as a function of temperature. We analysed the ultrafast transient in the DT signal as a function of the sample temperature, as reported in the newly added Fig. S12, Supporting information. The initial ultrafast dynamics of DT signal at cryogenic temperatures is found to be comparable in magnitude and dynamics to what measured at room temperature. The measurements demonstrate that the spontaneous polaronic deformation is not temperature activated, which may be related to the absence - or a very low value - of an energy barrier between the excitonic and polaronic states. We have now included an additional discussion and we believe that the additional measurements have significantly enhanced the manuscript insight.

13. Do the authors attempt to improve the quality of the polycrystalline thin film? According to the methods part, utilizing DMSO to dissolve the precursors is rare for iodine 2D perovskite. Meanwhile, the spin coating and annealing parameters are also critical to the film quality. Comparing single crystals and polycrystals is unfair when one is in poor condition. Also, the PMMA treatment could potentially affect the characterization. How did the authors evaluate the impacts?

Various sample preparation techniques have been explored, and we have now revised the manuscript to account for it. Our choice for the solvent can be supported by a comparison between the performance of thin films obtained from DMSO and DMF. If we consider the photoluminescence lifetime (in the low excitation regime) as an indicator of the quality of the thin film, we found that in the considered excitation range (below $1\mu\text{J}/\text{cm}^2/\text{pulse}$) samples prepared with DMSO have a PL decay around 200 ps, compatible with what reported in literature. Lifetimes for films prepared with DMF are substantially shorter, as reported in Fig. S14, Supporting Information. On the other hand, PL spectra are not noticeably affected by the chosen solvent.

Regarding the concerns about PMMA, we also compared the quality and reliability of the samples with and without PMMA deposition. We thus added in Fig. S14, Supporting information, a comparison between films with and without a PMMA protective capping layer. The TA spectra and lifetimes are unaltered, while we experienced a faster degradation of the sample without PMMA that, at highest excitation power, can become visible during the TA measurement (typically 15-30 minutes). Thin films covered with PMMA show better performance comparing PL lifetime at the same excitation energy, and have a better stability, allowing to perform multiple measurements without evident signs of alteration in the considered excitation range (below $10\mu\text{J}/\text{cm}^2/\text{p}$ at 430nm excitation wavelength).

14. How does halide perovskite superlattice (Nature volume 608, pages 317–323 (2022)) compare with bulk single crystals and polycrystalline thin films in the exciton stability?

Perovskite superlattice is indeed an interesting material, characterized by a lower exciton binding energy, with respect to the case of polycrystalline material or conventionally grown single crystals, showing a much higher mobility as well as higher and less-power dependent carrier lifetime. We did not explicitly consider superlattices in our study, so our observation may be speculative, still we can suggest that, since excitons are proven to be unstable “regardless” of morphology, we may expect superlattice material to have a behaviour analogous to what is reported in this work. We realize this material is worth a mention, thus we added a reference in the introduction of the manuscript.

15. The image quality of the main text should be improved.

We thank the reviewer for the note, we now uploaded new higher resolution images in the main text in place of the previous ones.

Reviewer #2 (Remarks to the Author):

The manuscript describes the results of the pump probe measurements of the 2D perovskites, namely PEA₂PbI₄ in n=1 Ruddlesden-Popper phase, both in single crystal and thin film form. Resonant excitation was used to create cold excitons. The main claim of the manuscript is exciton dissociation in 2D Ruddlesden-Popper layered perovskites, which would make them similar to their 3D counterparts. This is based on the observed splitting into free carriers within 1-2 ps as a consequence of the formation of polarons.

The experimental data are interesting, however I do not find that they support the final claim of the manuscript.

1. Why the splitting of the exciton signal in pump probe cannot be attributed to the formation of biexciton? For this polarization resolved measurements should be performed.

We thank the referee for carefully reading our manuscript, and considering mechanisms that were not explicitly discussed in the submitted version. Two are the basic reasons for which the formation of biexcitons (BX) must be ruled out. 1) Emission of one photon from a BX state would occur at the exciton (X) energy minus the BX binding energy. We never observed such emission. Furthermore, BX formation requires pairing of two Xs. BX emission should therefore overcome X PL at high excitation fluence. Again, experimental evidence of such a behaviour was never observed. 2) The fact that the transient PL intensity is proportional to the square of the transient DT signal is not consistent with the formation of BXs. If a population of BX existed, two cases would be possible. BXs are the minority species and Xs the majority one. In this case, the overall PL intensity would be proportional to the overall population of photoexcitations (i.e., Xs plus BXs), so would be DT. In other words, the PL transient would be proportional to DT, and not to DT squared as observed experimentally. If Xs were the minority species and BXs the majority ones, same conclusions are drawn.

However, we also believe that polarization-resolved measurements provide a direct and useful proof that biexcitons are not involved. Therefore we performed on thin film and single crystal samples additional DT and PL measurements with the control of circular polarization of both pump and probe. The results, that we report in Fig. S7, Supporting Information, show that there is no evidence of an effect on the DT signal or PL with pump and probe polarization, further excluding a biexciton explanation for the observed spectra.

We remark that such findings are not in disagreement with other measurements at liquid He temperatures that have shown biexciton formation, since at such very low temperature, excitons may very well be stable and then pair to form biexcitons. We also added a discussion and a relevant reference in the main text.

2. What are physical evidences of the polaron formation? For this either PL on a very short time scale should be measured (ACS Nano 2022, 16, 12, 21259–21265) or enhancement of the effective mass of the carrier, a characteristic sign of the polaron should be provided.

The main finding in our work is that excitons are unstable and dissociate, notwithstanding their very large binding energy. As a mechanism responsible for such a dissociation, we invoke the only one that is known to play a role, namely polaron formation. Such explanation is consistent with all our observations, time resolved DT spectra and DT vs PL decays; ultrafast transient absorption has been similarly interpreted in literature. Femtosecond PL has also been measured, but it provided no further direct evidence.

However, we do not claim to observe directly polaron formation. Ultrafast X-ray and electron diffraction measurements are the tools recognized as useful for providing direct evidence for polaron formation, as shown in pioneering experiments [Bao, D. et al. PRX ENERGY 2, 013001 (2023)] [B. Guzelturk et al, Nat. Mat., VOL 20, pp 618–623, 2021]. Their sensitivity is however still limited and such measurements

cannot access the whole range of fluences as DT and PL. In any case, such measurements are well beyond the scope of the present work and cannot be easily integrated with the resonant pumping of cold excitons.

3. At the same time I do not understand the physical process proposed by authors, is it polaron exciton formation? or is it dissociation and then carrier polaron formation? To which modes in each scenario there will coupling. This has to be quantified.

Our measurements demonstrate that excitons dissociate, therefore the formation of a majority of exciton-polarons are ruled out, because they would still be bound electron-hole states. The mechanism proposed to explain the observed dynamics is therefore exciton dissociation into unbound polarons, meaning that one exciton generates one positive and one negative polaron.

On the other hand, bound exciton-polarons have been reported in literature, especially through measurements at liquid helium temperatures (see eg. [Thouin et al, Nat. Materials 18, pp 349–356, 2019]). Such observations are not in contrast with ours, on the contrary it is perfectly reasonable that if the polaronic deformation is strong enough to dissociate the exciton at liquid nitrogen temperature, even if the binding energy is hundreds of meV, then some significant polaronic deformation persists even at liquid helium temperatures, when the exciton is stable against dissociation, giving rise to exciton polarons.

Stimulated by the Referee's comment, we therefore introduced in the main text the possibility that, when equilibrium is established, it may involve a majority of unbound polarons and a minority of exciton-polarons. Quantifying their ratio certainly warrants further investigation and may require applying the radiometric time-resolved photoluminescence we have recently demonstrated [Simbula et al. Adv. Optical Mater. 2021, 2100295].

4. Temperature dependent measurements should be provided, as coupling with the lattice should have a characteristic temperature dependence.

We welcome the Referee's suggestion and we have performed additional DT measurements at low temperature. Part of the answer to this question is bound to questions 2 and 3, and question 11 of Referee 1. We thus refer to the new data reported in Fig. S2, Supporting Information, showing DT on a picosecond timescale and its oscillations. Another interesting evidence is reported in Fig. S12, Supporting Information, that compares the DT signal evolution at room and at cryogenic temperature, showing that the dynamic is not affected in the sharp drop of intensity, that happens on the same timescale at the different temperatures. This indicates that the mechanism of exciton dissociation is quite not evidently dependent on temperature, for the considered temperature range.

5. What would be the mechanism of the dissociation of exciton with binding energy of few hundred meV? Why would polaron dissociate the exciton?

The Referee's comment highlights the major novelty of our findings: the experimental evidence demonstrates that excitons dissociate, even though their binding energy estimated from absorption measurements is several hundreds of meV, at least an order of magnitude larger than thermal energy kT at room temperature. The mechanism we invoke to explain the observation is that the polaron stabilization energy, i.e. the energy gained through lattice relaxation in presence of optical excitations, is comparable to the exciton binding energy. Therefore, the energy difference between bound excitons and unbound polarons is not the exciton binding energy anymore, and is instead the difference between the exciton binding energy and the polaron relaxation energy. Such mechanism is explained in detail,

together with the modified Saha equilibrium condition and a rate equation model that describes the experimental observations, in Simbula et al. *Adv. Optical Mater.* 2021, 2100295.

Following the Referee's comment, we therefore modified the discussion in the manuscript and provided a less ambiguous and more clear description of the invoked mechanism to dissociate the exciton.

6. Just bimolecular decay is not sufficient to claim charge separation, as in 2D TMDS, where we have strongly bound exciton and still Auger effect is observed.

The Referee is indeed right, bimolecular decay alone is not sufficient to draw conclusions on exciton dissociation. This is the reason why we combine PL and DT in a "tandem" setup where PL and DT are measured on the same spot of the sample, and show that PL is proportional to the square of DT – as reported in Fig. 4 and Fig. S5, Supporting information. Such evidence is conclusive for charge separation and for the fact that an equilibrium is established between a majority of unbound carriers (the polarons) and a minority of bound excitons. The two different power laws of the $d(\text{PL})/dt$ vs PL and $d(\text{DT})/dt$ vs DT identify the dominant decay process for polarons and excitons as the monomolecular decay of excitons. Because of Saha equilibrium, such a decay results in a bimolecular depletion of polarons. Auger would instead result in a cubic rate for polarons and a quadratic one for excitons, as experimentally observed and discussed in Simbula et al. *Adv. Optical Mater.* 2021, 2100295.

Following the Referee's comment, we improved the discussion of the implication of our measurements to clarify any possible misunderstanding.

Before providing this extra data and answer to above question, the manuscript is not suitable for publication.

Reviewer #3 (Remarks to the Author):

The manuscript "Exciton splitting in 2D layered metal-halide perovskites" presents a smart experimental study of dynamics of dissociation of bound excitons in a two-dimensional (Phenethylammonium)₂PbI₄ perovskite using time-resolved femtosecond differential transmission techniques. The results indicating their dissociation into free carriers within 1-2 ps can be important for understanding of nature of exciton dynamics in 2D Ruddlesden-Popper layered perovskites and implementation of the materials into photovoltaic devices.

Unfortunately, the authors use some jargon in the manuscript, such as "exciton splitting" and "excitons form and then split" which obscures the physics of the processes taking place. It is difficult to imagine that exciton split something or be spatially split in the material. Probably, a splitting of their energy states is meant. In this case, it would be better if the authors used more precise terminology. It also remains unclear from the manuscript the value of this splitting. Does it correspond to the wavelength distance between the maximum and the minima in Fig. 2e? In order for the reader to better perceive the processes that the authors use to explain the experimental results, it is absolutely necessary to give an energy scheme with an indication of transitions on it at different excitations used by the authors in the experiment.

We sincerely thank the Referee for pointing out a source of possible misunderstanding in the terminology and phrasing we have been using. We have therefore thoroughly revised the manuscript, including its very title, and avoided the wording 'exciton splitting' that generated some ambiguity to the Referee, since we do not describe any energy splitting. In place, we refer to the process as 'exciton

dissociation', meaning that a bound exciton dissociates into one positive and one negatively charged carrier, holes, and electrons, or more accurately positive and negative polarons, that are not bound to each other.

After revising every instance in the manuscript where we refer to the process and carefully avoiding sources of possible confusion, we believe the manuscript is now compliant with the broadly adopted terminology and avoid conflicts with other definitions in literature [Han, Y. et al. Nat. Mater. 21, 1282–1289 (2022)].

Other comments:

Specifying $n = 1$ in the abstract is not clear and requires an explanation. "PEA": using abbreviation in the abstract is not a good style.

We thank the referee for the comment. We now replaced specified what PEA stands for and removed the $n=1$ from the abstract, while we included a more precise definition for it in the main text.

Section Material and methods, first paragraph:

" $n = 1$ " is still puzzling, what is n here? A physical value or a number? If latter, in what sequence?

Can authors characterize the phase they mention here by more informative terms than " $n = 1$ "?

Thanks to this point raised by the referee, we realized that we did not give an exhaustive definition of the crystalline structure (i.e., 2D Ruddlesden-Popper $n=1$ phase). To make the manuscript more comprehensible, even in the perspective of reaching a broad audience, we added a statement where we describe more carefully the structure of the analysed material.

Resolution of Fig. 1a needs to be improved.

We thank the referee for the notation, compatible with that of other referees. We now replaced Fig. 1, as well as other figures in the main text, with other ones with higher resolution.

Reviewers' Comments:

Reviewer #1:

Remarks to the Author:

This revised manuscript answered the reviewer's comments well and made it complete. This manuscript now meets the publishing standard of the journal.

Reviewer #2:

Remarks to the Author:

First of all I would like to thank the authors for a detailed reply to questions and comments raised by referees. However, I am still not convinced if the interpretation of the authors is correct. Namely, the interpretation of the dissociation of the exciton, despite its large exciton binding energy. The dissociation of the exciton into carriers and possible formation of the polaron would indicate absorption/emission in very different energy than for exciton (much closer to the band gap). This is not the case, especially that the emission from 2D perovskites is really observed close to the energy of the 1s exciton transition, and the emission is very strong and efficient which would not be the case if the exciton dissociate. Therefore, and also based on comments on the authors that they really do not have strong evidences of the exciton dissociation and further polaron formation. Hence, I do not recommend the manuscript for publication in the present form as the interpretation of the data is not supported by experimental results.

Reviewer #3:

Remarks to the Author:

I am satisfied with the author's decision to change the terminology to a more understandable one. However, I didn't notice that the authors added a diagram of the process they describe, which I recommended. I still believe that such a scheme (whether an energy diagram, illustration of spatial motion or a mixed picture), is tempting here to facilitate the understanding of the essence of the phenomenon to the inexperienced reader in this matter.

Point-by-point response to the reviewers' comments

Reviewer #1 (Remarks to the Author):

This revised manuscript answered the reviewer's comments well and made it complete. This manuscript now meets the publishing standard of the journal.

We are pleased that Reviewer #1 is satisfied with our revision of the manuscript.

Reviewer #2 (Remarks to the Author):

First of all I would like to thank the authors for a detailed reply to questions and comments raised by referees. However, I am still not convinced if the interpretation of the authors is correct. Namely, the interpretation of the dissociation of the exciton, despite its large exciton binding energy.

In the first report, Reviewer #2 suggested biexciton dynamics and Auger recombination as possible explanations for our experimental results. We very much appreciate that these alternative explanations are no longer part of the discussion after our reply, which also included the results of the additional experiments she/he requested. However, Reviewer #2 is still not convinced by our interpretation of the experimental data in terms of exciton dissociation and raises these further new objections:

The dissociation of the exciton into carriers and possible formation of the polaron would indicate absorption/emission in very different energy than for exciton (much closer to the band gap). This is not the case, specially that the emission from 2D perovskites it is really observed close to the energy of the 1s exciton transition, and the emission is very strong and efficient which would not be the case if the excitons dissociate. Therefore, and also based on comments on the authors that they really do not have strong evidence of the exciton dissociation and further polaron formation. Hence, I do not recommend the manuscript for publication in the present form as the interpretation of the data is not supported by experimental results.

First, we stress that we agree with the Referee on the fact that light emission results from radiative recombination of excitons, as actually reported in the main text: "To identify the emitting species, we can note that the PL spectrum is always resonant with the exciton absorption line; therefore, optical emission should always be ascribed to excitons, even if they were the minority species,^{34,69} with the PL signal proportional to the exciton density n_X ."

However, we disagree with the Reviewer's conclusion that since light emission results from radiative exciton recombination, excitons do not dissociate. This conclusion would imply that, under resonant excitation, excitons are the majority species at all times after excitation. Indeed, the experimental results (Figure 6 and Figures S4 and S5) clearly show that once equilibrium is established, excitons are the minority species, and the species into which they dynamically dissociate is the majority one. In other words, if the Reviewer's belief were true, PL and DT would both be proxies of the exciton population, resulting in a proportionality relation between PL(t) and DT(t), whereas the

experimental results clearly showed that PL(t) scales as DT(t)², even at low temperature (see Figures S4 and S5), in agreement with our picture of the radiative recombination pathway.

To better clarify our point, also accordingly to the request of Reviewer #3, we decided to add Figure 6 in the discussion paragraph, showing a sketch of the involved optical transitions and excited-state processes taking place. We also explicitly introduced the equation for the kinetics reaction that leads to the equilibrium between excitons and polaron pairs: $p^+ + p^- \leftrightarrow X$, and light emission following radiative recombination of excitons: $X \rightarrow \hbar\omega_X$ (see equation 2 in the main text). In the previous work *Adv. Optical Mater.* 2021, 2100295 [Ref. 34], the relevant rate equations and modified Saha's equilibrium constants were derived, while in the paper *Energy Environ. Sci.*, 2022, 15, 1211 [Ref 30], we provided a direct quantitative evaluation of the effect of the presence of unbound carriers on the radiative quantum yield. In this paper we showed that, in equilibrium conditions, efficient light emission is fully consistent with exciton dissociation.

Are charged carriers (i.e., polarons) the majority species into which excitons dissociate? We have provided strong evidence in favour of this hypothesis by analysing the nonlinear response of the exciton absorption resonance through the DT spectrograms and contrasting it with what we expect to find for the nonlinear exciton response to a population of (uncharged) excitons and a population of charged carriers. Figures 2-4 clearly support the interpretation of charged carriers as the majority species into which excitons dissociate. Few weeks ago, the work 'Exciton Formation Dynamics and Band-Like Free Charge-Carrier Transport in 2D Metal Halide Perovskite Semiconductors' by the Herz et al. was published on *Advanced Functional Materials*. The Authors have monitored the formation of charge carriers by time-resolved visible pump - THz probe spectroscopy and their conclusions are consistent with those reported in our manuscript, namely that despite exciton binding energies in excess of 200 meV, an abundant charged carrier population is formed, even at low temperatures.

We understand the reluctance of Reviewer #2 to accept that, in semiconductors with exciton binding energies well above the thermal energy, the majority of photoexcitations are free carriers. However, the emission properties and nonlinear optical response of 2D metal halide perovskites clearly show that 2D perovskites have the unique property of being free carrier semiconductors, despite large exciton binding energies. The microscopic mechanisms of interaction between carriers and the lattice that favour exciton dissociation remain to be investigated from a theoretical point of view. However, this is beyond the scope of the present manuscript.

Reviewer #3 (Remarks to the Author):

I am satisfied with the author's' decision to change the terminology to a more understandable one. However, I didn't notice that the authors added a diagram of the process they describe, which I recommended. I still believe that such a scheme (whether an energy diagram, illustration of spatial motion or a mixed picture), is tempting here to facilitate the understanding of the essence of the phenomenon to the inexperienced reader in this matter.

We thank Reviewer #3 for the suggestion, and we also agree that our discussion could gain in clarity if supported by an illustration. We now added Fig. 6 in the main text, showing a sketch of the relevant photophysical and equilibrium excited-state processes occurring in the system.